# The Journey to Making 'Digital Technology' Education a Community Learning Venture

Fiona Carroll * , Rumana Faruque, Chaminda Hewage , Vibhushinie Bentotahewa and Sophie Meace

Cardiff School of Technologies, Cardiff Met University, Llandaff Campus, Western Avenue, Cardiff CF5 2YB, UK; rfaruque@cardiffmet.ac.uk (R.F.); chewage@cardiffmet.ac.uk (C.H.); vibentotahewa@cardiffmet.ac.uk (V.B.); smeace@cardiffmet.ac.uk (S.M.)
* Correspondence: fcarroll@cardiffmet.ac.uk; Tel.: +44-0781-5065-514

**Abstract:** Technology has become an integral part of our educational systems, and its importance in our schools cannot be overstated. However, digital skills, unlike other literacy skills, such as reading, writing, and numeracy, still have many discontinuities between how children use them at home versus in school. Therefore, in Wales (UK), digital skills are being promoted as part of the Digital Competence Framework (DCF) and feature highly in the new Curriculum for Wales (2022). Moreover, the new Digital Technology General Certificate of Secondary Education (GCSE) in Wales has been introduced to provide learners with the opportunity to gain a qualification that builds digital skills, knowledge, and understanding. However, this also brings many challenges for teachers, such as a lack of confidence, knowledge, and training, as well as a lack of resources and fear of change, to name a few. These challenges, in turn, have an impact on pupils' motivation and performance, as well as parents' ability to support their children. This paper presents a qualitative case study on the development of a new digital technology learning community for primary and secondary school pupils, their teachers, and parents in Blaenau Gwent, Wales (UK). Firstly, the paper will provide insight into what was required to establish an effective learning community, including ensuring engagement and buy-in from all stakeholders. Secondly, through the description, analysis, and interpretation of findings from two studies, the paper will highlight the impact of the DTLSN learning community on teachers and pupils in Blaenau Gwent, especially in terms of their learning and teaching.

**Keywords:** learning; community; digital competence



## 1. Introduction

Technology is perhaps the strongest factor shaping and moulding the educational landscape today [1]. Moreover, school curricula must evolve quickly to meet the growing demand for digital skills. With this comes many challenges, and as a result, many educators are turning to learning communities to grow in their craft [2]. It is not just teachers who are struggling to cope with the transition to more technology-related subjects and teaching practices, it is also the pupils they teach and the parents who want to support them. For example, in Wales, many schools are facing new challenges around the use of technology in their subject areas. Digital skills, which are part of the Digital Competence Framework (DCF) [3], feature highly in the new Curriculum for Wales 2022. Furthermore, the new Digital Technology GCSE [4] was introduced this year in Wales to provide learners with a broad range of digital skills, knowledge, and understanding. This is an inspiring step forward by the Welsh Government, but at the same time, it can be a lot for a busy teacher, pupil, or even a parent to keep on top of. The authors of this paper provide a qualitative case study to highlight that learning communities can be an effective way to fill the gap and support teachers, pupils, and parents as they attempt to keep abreast of the new digital technology skills required in and out of the classroom.

Indeed, teachers know the value of creating small learning communities within their classrooms, especially for engagement and collaborative learning. As Walker et al. ([5], p. 1) note, 'Learning circles are an enabling process to critically examine and reflect on practices with the purpose of promoting individual and organizational growth and change'. This paper presents two studies that aim to explore teachers' and school pupils' impressions of a new digital learning community (Digital Technology Learning Support Network—DTLSN) in Blaenau Gwent, Wales (UK). The following sections of this paper will discuss the Digital Competence Framework and the new Digital Technology GCSE subject in Wales. It will emphasize the need for a learning community to support teachers and pupils with digital technology. Importantly, the paper will discuss what was required to set up the DTLSN. Through the findings from two studies, it will highlight the positive impact of such a learning community on teachers and school pupils in their ever-increasing technological-driven educational system.

## 2. Background: The Digital Technology Landscape in Wales (UK)

The mandatory cross-curricular skills within the Curriculum for Wales are literacy, numeracy, and digital competence. It is expected that all practitioners, across all curriculum areas, will create opportunities to develop and ensure progression in these skills. To support educational establishments and educators in developing and applying these skills, the Literacy and Numeracy Framework (LNF) [6] and the Digital Competence Framework (DCF) [3] have been used as non-statutory supporting guidance. The DCF in Wales aims to promote competencies in digital citizenship, interaction & collaboration, production, and data & computational thinking. The aim is to enable citizens in Wales to live with the ever-changing digital landscape and contribute to the growth of the IT sector.

In more detail, under the ethos of citizenship, learners will enhance their ability to contribute positively to the digital ecosystem and critically evaluate their role and place within the digital world. Citizenship reinforces identity, image, reputation, health and well-being, digital rights, licensing and ownership, online behavior, and online bullying. Effective and secure communication, collaboration, storing, and sharing are also key learning aspects of the interaction and collaboration ethos. The production element covers sourcing, searching, planning digital content, creating digital content using a diverse range of software, and evaluating and improving digital content. This is an important skill where content creation remains a key aspect of the present society. The data and computational thinking element highlight the importance of computational thinking, which combines scientific inquiry, problem-solving, and thinking skills. The key elements are problem-solving and modeling, as well as data and information literacy. These enhance the learner's ability to create and refine algorithms and flowcharts to solve problems. Furthermore, this supports the construction, refinement, and interrogation of datasets within tables, charts, spreadsheets, and databases to test or support an investigation.

A clear delivery plan has been laid out by the Welsh Government to embed digital competence as a cross-curriculum responsibility. Moreover, since September 2022, this has been available. The following are the four main support mechanisms for practitioners to plan and deliver DCF:

- Welsh Government resources (classroom task ideas, professional learning needs, and mapping tools, animated explainers on the DCF).
- Four regional consortia (hosting events and offering advice and support for schools).
- Estyn (identification and sharing of any good practice observed in schools in preparing for the DCF).
- Digital pioneer schools.

Despite the above-pledged support mechanisms by the Welsh Government, the perceived 'challenges' of practitioners are complex and far-reaching, including resistance to change, lack of professional training and/or Welsh Government guidance, time and resource allocation, fear of 'getting it wrong,' lack of clarity regarding assessment, and the potential for patchy provision on a national level (with urban/rural disparities). Moreover,

while expressing levels of optimism and enthusiasm about the new curriculum, many teachers indicated high levels of anxiety about its practical application. In seeking out the 'new' within the new curriculum, it seems that entrenched practices may limit the potential for transformation. Similar to the more traditional types of literacy (e.g., reading, writing, and numeracy), digital competence development should start in the home and further grow in and through the school's life cycle. To achieve this, teachers, pupils, and parents need more hands-on support.

## 3. The Learning Community

Learning communities can provide a space and a structure for people to share common goals and learn together. As Cross ([7], p. 4) defines, they are 'groups of people engaged in intellectual interaction for the purpose of learning.' Despite learning communities having been approached from different perspectives that serve different purposes [8], they have certainly been noted to enhance learning, empower students, and increase engagement in the learning process [9]. Indeed, these communities can be both aspirational and inspirational in how they enable people to connect, share ideas, and learn from one another. As Shea et al. [10] highlight, the community can play a critical role in building and sustaining productive learning. Moreover, Lieberman and Mace [11] describe the importance of sharing accomplished practices between classrooms and practitioners with varying levels of experience. Learning communities are important in how they bring together teachers, pupils, parents, etc., to support one another in their learning through shared knowledge and experience.

Furthermore, Shea et al. [10] emphasize that teaching presence is among the most promising mechanisms for developing online learning communities. The teacher is key in ensuring that pupils feel motivated to learn and can also learn themselves. As Prenger et al. ([12], p. 1) note, 'Professional learning communities are promising for teacher learning and improving the quality of education.' Teachers can learn new teaching skills while boosting their confidence and energizing their own teaching and learning. It has been found that professional learning communities can help teachers develop the curriculum, support students' understanding of development, overcome learning difficulties faced by students, and make learning designs more effective and efficient [13].

*What Makes an Effective Learning Community?*

The authors of this paper believe that all stakeholders in the learning experience should have opportunities to learn and develop their practices through collaboration with others. Tarnanen et al. [14] believe that teachers' professional development and learning (PDL) is an important part of supporting students' learning, i.e., in-service teacher education that meets teachers' individual needs and inspires them to develop their teaching practices while providing opportunities for collaboration and shared learning.

For teachers to respond to curriculum changes and develop their own teaching practices, they need 'multiple opportunities to learn new information and understand its implication for practice' ([15], p. 15). In recent years, collaborative networks have proven to be an effective way of supporting professional learning and have the possibility to drive educational change [16,17]. Collaboration is defined as 'the process of sharing learning and engaging in dialogue to create new learning that informs future actions' ([18], p. 8). A recent study in Wales [19] revealed that school-led initiatives, such as digital working groups, digital libraries, and digital champion schemes have a significant positive impact on developing teaching practices with technology. Participants in the study stressed the importance of how being able to connect and collaborate provided a safe space for them to share good practice ideas and learn from each other, supporting them in building skills and confidence in implementing the DCF in Wales.

The fully online learning community (FOLC) model that was introduced by [20] supports some of the key elements of this transformational learning. It is about facilitating richly collaborative, socially cohesive, and constructively critical learning communities

supported by a flexible array of synchronous and asynchronous digital affordances [20]. Wenger ([21], p. 1) further stresses that 'the success of organizations depends on their ability to design themselves as social learning systems...'. She goes on to describe four important disciplines of the learning partnership ([21], p. 12):

- The discipline of the domain (e.g., what is the learning partnership about?)
- The discipline of the community (e.g., who needs to be involved so the partnership can make progress?)
- The discipline of practice (e.g., what should participants do together to learn and benefit from the partnership? etc.)
- The discipline of convening (e.g., who will take leadership in holding a social learning space for this partnership?)

In the book, The Fifth Discipline: The Art and Practice of the Learning Organization, Senge [22] identifies five interdependent core disciplines of the learning organisation, i.e., personal mastery, mental models, building a shared vision, team learning, and systems thinking [22]. As Cochrane et al. [23] highlight in the building of their global community of learning, there are four stages—stage 1: establishing a core community of practice (CoP), stage 2: brokering participation; stage 3: nurturing participation; stage 4: brokering practice. They note the need to utilise mobile social media for scaffolding and sustaining interaction in the global CoP across all four stages [23]. Indeed, it is very much about how the learning communities connect with the stakeholders, how they are developed, and how they are supported. This can come down to having the right structures in place, the flexibility of the stakeholders to build relationships, the right content, the mode of interaction, having communication in place, etc. The following section details the journey of setting up the Digital Technology Learning Support Network (DTLSN), which is a learning community designed to enhance the digital technology learning experience for the community of teachers, pupils, and parents.

## 4. The DTLSN Project and the UKCRF

The Digital Technology Learning Support Network (DTLSN) is a project funded by the UK Community Renewal Fund (UKCRF) that ran a testbed learning community in Blaenau Gwent, Wales (UK) from January to July 2022. DTLSN aims to provide an innovative learning experience that involves the whole community, teaching children, parents, and teachers together. In Blaenau Gwent, the project involved Cardiff Metropolitan University (CMU) in conjunction with the Technocamps Programme (TC), Swansea University, and ten Welsh primary and two secondary schools (including teachers, pupils, and parents). The network delivered over thirty-eight innovative learning workshops for the schools, offered one-to-one support sessions for the teachers, and ran online talks and taster sessions for parents. These workshops involved over five hundred and thirty-one pupils, over fifteen teachers, and seven parents. For primary schools, the DTLSN project provided at least two classes per school with three innovative workshops (approximately two hours each). The first workshop focused on computational thinking. The second workshop offered a choice of one of three machine learning workshops (i.e., modelling zombies, butterfly hunter, and the white feather). The third workshop was about implementing a digital technology solution using the Robotics Lego Spike Prime kit (see Figure 1). Two career-fit and Python programming workshops were run in the secondary schools. This test-bed project was piloted in the Blaenau Gwent (BG) region, and the goal was to provide cohesive learning support for all stakeholders. The following section provides a snapshot of what was required to set up the DTLSN learning community.

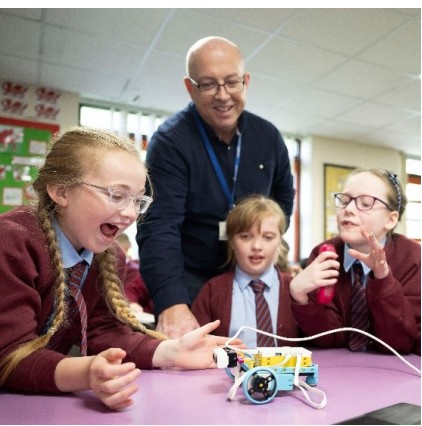

**Figure 1.** Robotics Lego Spike Prime kit workshop.

*Building a 'Digital Technology' Learning Community*

DTLSN required a community that would support a learning partnership [21], and in doing so, would encourage and promote digital technology thinking and learning (see Table 1). To achieve this, a learning community identity/presence was created. Firstly, a DTLSN brand was designed (e.g., the logo (see Figure 2)). Secondly, the foundation of a strong communications channel was established. This was in the form of a social media backbone (e.g., public-facing DTLSN Twitter, Facebook, LinkedIn, and Instagram pages were created) along with a private Microsoft Teams site for the schools. These came together to form the core communication channels for the community. In detail, they gave the stakeholders the ability to connect with one another, providing a sense of collective identity. As the research shows, social media can be used to highlight and share the best practice in a learning community as well as access new and shared relational resources (attitudes, knowledge, ideas, practices, etc.) [24]. As we can see evidenced in Figure 3, it was important for the DTLSN community momentum (and motivation) to showcase the work undertaken in the school workshops across these channels. It allowed for visual sharing of the best practices.

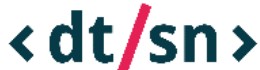

**Figure 2.** DTLSN logos.

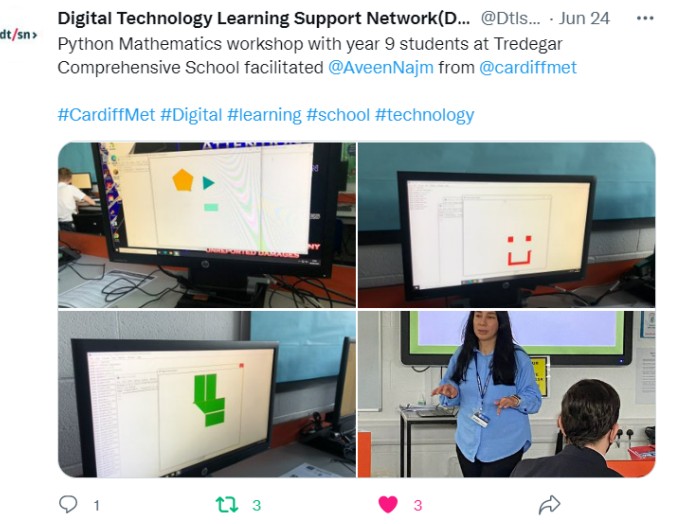

**Figure 3.** Using Twitter to share what is happening in the workshops.

**Table 1.** Four disciplines of the DTLSN learning partnership.

| Domain | Community | Practice | Convening |
|---|---|---|---|
| An innovative learning experience that works with the whole community. | School pupils, parents, and teachers. | Digital Technology workshops, such as computational thinking and Lego workshops | Local authority and Welsh Government |

Once the social media infrastructure was in place, the DTLSN team focused on building up the learning community content. As mentioned previously, it was about setting up the foundations to facilitate a richly collaborative, socially cohesive learning community. To achieve this, it was believed that teachers, pupils, and parents should be at the heart of the community, so engagement with these stakeholders was key. The focus was on designing and delivering appropriate and effective learning and teaching experiences. In detail, the DTLSN learning community needed to be something that the stakeholders could be a part of and also where their learning needs could be understood and addressed. Therefore, innovative and creative school workshops were provided to engage both the pupils and teachers in digital technology learning and the application of new digital technology skills. Other opportunities were also explored to ensure that all stakeholders could feel part of the learning community. These included running competitions for the school children (see Figure 4), technology-inspired days out at the university for pupils and their teachers (see Figure 5), and finally, a celebration event to bring everyone together (see Figure 6) to celebrate their digital technology learning. In terms of the parents, the DTLSN learning community also ran online sessions to cater to more top-level learning and awareness needs around digital skills and digital technology usage (see Figure 7).

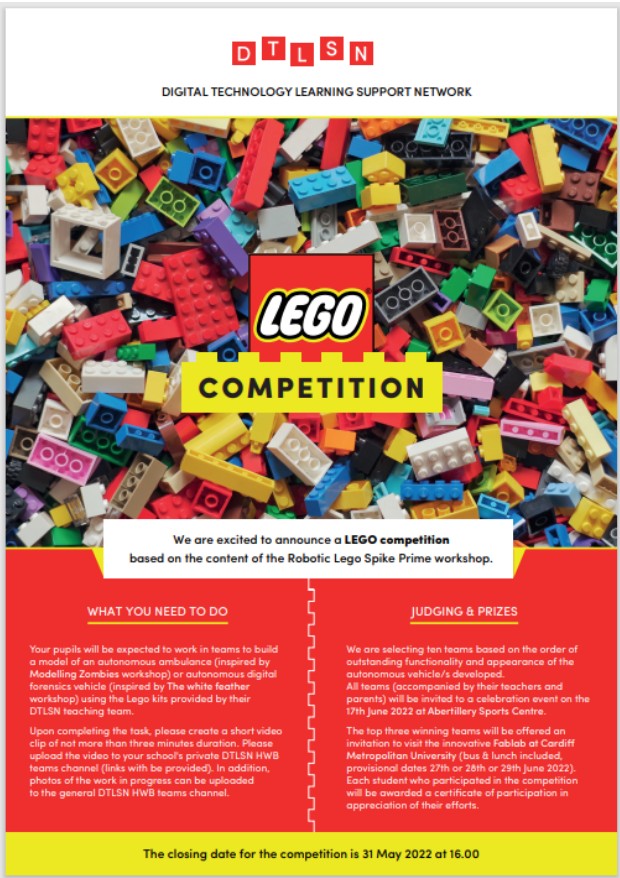

**Figure 4.** Lego competition poster.

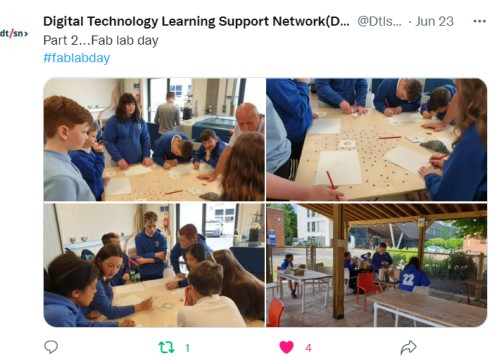

**Figure 5.** Day-out at the university.

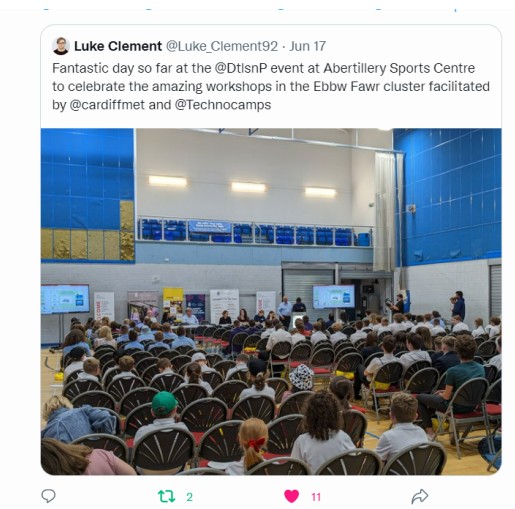

**Figure 6.** Celebration event.

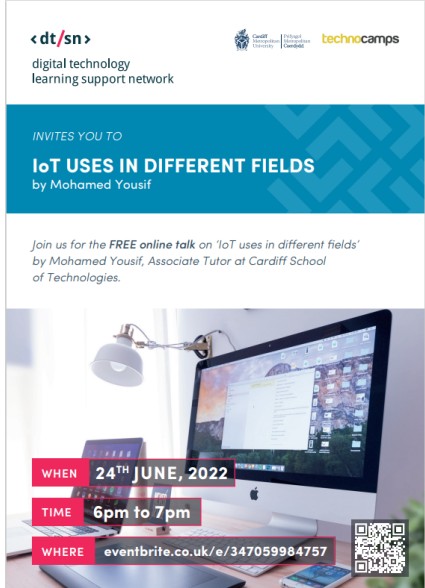

**Figure 7.** Online session for parental learners.

## 5. Research Studies

The research investigated the effect of the Digital Technology Learning Support Network on the learning community in the Blaenau Gwent (BG) region. While most schools in the region were invited to participate in DTLSN workshops, only twelve schools chose

to do so. Two studies were conducted in these twelve schools (ten primary and two secondary) between April and June 2022, with the approval of the ethics committee at Cardiff Metropolitan University.

*5.1. Study A*

The first study aimed to provide insight into the impressions of twelve teachers regarding the value of the DTLSN learning community and its workshops.

5.1.1. Research Methodology

A survey research method was used to collect data from the participants. A questionnaire was designed to gauge the teachers' initial responses to the DTLSN. The questionnaire consisted of twenty-five questions, taking approximately twenty minutes to complete, and included multiple-choice and text-entry questions. The questionnaire was divided into four sections, with the first section focusing on the teachers' background, their teaching subject, and the ICT/computing setup of their school.

- How long have you been teaching?
- Do you have a degree-level qualification (or equivalent) in computer science, IT, or a related subject?
- Which level of curriculum do you teach?
- Which qualifications do you teach?
- Do you believe that changing the ICT/ Computing curriculum has provided benefits to pupils?
- How confident are you about the new Digital Technology curriculum?
- How has your school adapted its ICT/Computing provision (timetabled hours) in relation to the new curriculum?

The next section focused on the participant's DTLSN experience. The following questions were asked:

- Which DTLSN services have you participated in?
- How would you rate the quality of the DTLSN services you have participated in?
- How would you rate the quality of support for pupils who attended the DTLSN school workshop?
- Overall, how would you rate the support provided by DTLSN?
- What impact do you believe DTLSN has had on the following groups: Teachers, pupils, and the school?
- What skills have the following people gained after they participated in DTLSN: Yourself, pupils?

The third section of the questionnaire focused on the teachers' opinions on the skills gap and how they felt the DTLSN would support and help to address the gap.

- What are the barriers that you personally have faced in skills development?
- Do you think the support provided by DTLSN could be delivered by your school if DTLSN did not exist?
- Following the completion of the DTLSN project, would you like to receive additional support of a similar nature?

The final section of the questionnaire focused on analysing the DTLSN to determine where it could be further enhanced.

- What additional support do you believe DTLSN could provide to pupils to further their interest in STEM subjects?
- would sustaining the DTLSN project impact the following groups: Yourself, pupils, teachers, and school?
- Please rate how the DTLSN operation met your expectations:
- Do you see any notable strengths of DTLSN?
- Do you see any notable weaknesses of DTLSN?

- Do you see any notable opportunities for DTLSN?
- Do you see any notable threats to DTLSN?
- What impact has COVID-19 had on your school, particularly in relation to the provision of ICT/computing lessons?
- What impact has COVID-19 had on the delivery of DTLSN support?

The questionnaire was printed out and distributed to the participants at the beginning of the workshops, with the aim to complete it during the session. This took place over a period of three months (April 2022 to June 2022). All participants were given the choice to participate in the research or decline. The analysis of the data collected for this study was assisted by Ciotek, Ltd.

### 5.1.2. Participants

Eleven teachers and one student teacher took part in the study. These participants ranged from a postgraduate certificate in education (PGCE) student to those with twenty-one-plus years of teaching experience. Ninety-two percent of the participants had at least six years of teaching experience, underlining the duration of experience within this field.

### 5.1.3. Findings

The findings reveal that none of the participants felt 'very' confident or 'not at all' confident in the technology-oriented new curriculum in Wales. As shown in Figure 8, four participants reported that they were not very confident, which was attributed to a 'lack of personal training'. However, the remaining eight participants reported feeling slightly confident for various reasons, including:

- STEM projects are helping me find my confidence. (P1)
- It'll take some time getting used to up-staging myself through training. (P2)
- I have not used coding for a long time, but workshops are helpful. (P3)
- Personal learning opportunities support development and confidence. (P4)
- I am eager to learn myself because pupils will engage better. (P6)
- Lack of personal training. (P8)

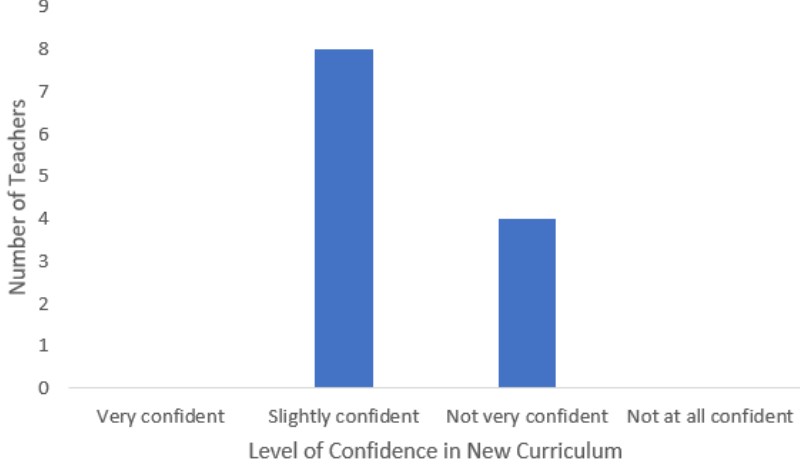

**Figure 8.** Level of confidence in the new curriculum.

Interestingly, as Table 2 highlights, several of the participants did not have an educational background in computer science, IT, or a related subject. Furthermore, many participants identified a number of barriers to skills development as demonstrated in Figure 9. This resulted in six participants expressing limited time availability, six participants feeling they had gaps in skills and knowledge, and four participants emphasizing limited resources.

**Table 2.** Do you have a degree-level qualification (or equivalent) in computer science, IT, or a related subject?

| Participant | Degree-Level Qualification |
| --- | --- |
| P1 | Yes—in a non-science-related subject |
| P2 | Yes—in a non-science-related subject |
| P3 | Yes—in a non-science-related subject |
| P4 | No |
| P5 | Yes, a higher national diploma in computer science |
| P6 | No |
| P7 | Yes—in a non-science-related subject |
| P8 | Yes—in a non-science-related subject |
| P9 | No |
| P10 | No |
| P11 | No |
| P12 | No |

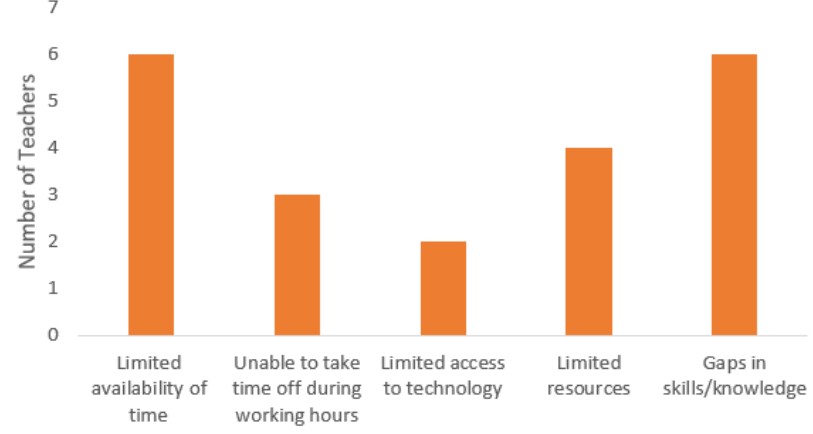

**Figure 9.** Barriers to skill development.

In terms of the DTLSN experience, many participants believed that the DTLSN had a positive impact on pupils, teachers, and schools. The following highlights some of the impact on pupils:

- More access to STEM and Digital Competence Framework resources and support. (P2)
- Consolidating knowledge and skills in the area of coding and digital technology. (P3)
- Raised enjoyment levels. (P4)
- Increased levels of critical thinking. (P5)
- Increased enjoyment, motivation, and engagement. (P6)
- Pupils have more confidence. (P8)

The participants felt that teachers, similar to them, benefited from the DTLSN in the following ways:

- Supported in the delivery of the curriculum. (P2)
- Consolidating knowledge and skills. (P3)
- Beneficial multi-modal learning delivery. (P4)
- Support gives teachers ideas and confidence to teach new skills to pupils. (P5)
- Increased confidence and ability to make links within topics. (P6)
- Better understanding of computational thinking. (P9)

Finally, they felt that the school benefited through the below:

- Great support across multiple-year groups with skill development. (P2)
- Delivered further support during staff meetings. (P3)
- Improved school experiences and offers for pupils. (P6)

When probed about the skills gap and whether they believed the support provided could be delivered by their school if DTLSN did not exist, 83% of participants provided a response. One participant believed that their school could deliver the provision, six participants did not think their school could deliver the provision, and three participants were unsure. The general consensus among participants was that they would need sufficient training, knowledge, and confidence, particularly relating to coding, to be able to deliver the provision without the support of the DTLSN learning community.

Of the ten participants who responded to this question, all confirmed that sustaining DTLSN would impact teachers and pupils; nine teachers advised that it would have an impact on schools. The following outcomes are suggested as a result of sustaining DTLSN:

- Can increase teachers' confidence, which could then allow them to deliver the programme to the whole school. (P1)
- Increased skills and knowledge development and curriculum coverage. (P2)
- DTLSN has raised pupils' aspirations and enjoyment. Additional support could further extend the development of teachers' and schools' understanding of the subject. (P4)

Table 3 highlights some participants' thoughts on extending the DTLSN.

**Table 3.** Do you see any notable opportunities for DTLSN?

| Participant | Opportunities |
|---|---|
| P1 | To teach further down the school (e.g., Yr 3, Yr 4) |
| P2 | Opportunities for further collaborations |
| P3 | To work with younger age groups rather than year 5, year 6 |

Finally, some participants expressed concerns about the speed of technological change and the need for a continual training programme for teachers. Others felt that the teaching of traditional subjects, such as history, geography, biology, mathematics, English, and foreign languages needed minimal ongoing development once the curriculum was established.

### 5.1.4. Discussion

Overall, it can be noted that the DTLSN learning community was of value as eight participants confirmed that the support for students was excellent, with four confirming it was good. The following reasons were cited:

- The excellent support provided to develop new skills and knowledge. (P1)
- Great resources and teaching, allowing children to fully engage. (P3)
- Interactive sessions that support understanding. (P4)
- Very experienced and friendly staff. (P5)
- Three members of staff on hand to support all pupils across the room. (P6)
- Great support given with enjoyable and engaging lessons. (P1)

Finally, participants were asked to rate how the DTLSN operation met their expectations. Of the participants who responded, four participants indicated that DTLSN had exceeded their expectations and five participants suggested DTLSN had met their expectations.

### 5.2. Study B

The second study investigated school pupils' thoughts and feelings about the DTLSN learning community, particularly the subject of computer science/digital technology.

### 5.2.1. Research Methodology

A short questionnaire (consisting of seven questions) was used to collect data from participants. It followed a standard format with both open-ended and closed questions. These questions mainly consisted of multiple-choice but some allowed for comments. It included questions such as 'Did you enjoy the activities and the equipment we used?' to 'Are you more interested in computer science/digital technology now?'. The questionnaire

was designed in a simple way to ensure that all age groups could easily understand what was being asked. The study took approximately ten minutes to complete. It was printed on one page and immediately delivered to the participants after each of the three workshops. This occurred over a three-month period (April 2022 to June 2022).

### 5.2.2. Participants

Over five hundred and thirty-one school pupils between the ages of eight years old and fourteen years old took part in this study. These participants partook in all DTLSN workshops, which were delivered to twelve schools in the Blaenau Gwent region, Wales, UK, in the spring and summer of 2022. To avoid sampling bias, one thousand one hundred participant questionnaires were randomly selected from a larger collection of one thousand five hundred and ninety-three participant questionnaires.

### 5.2.3. Findings

After each of the three workshops, the participants were asked their thoughts and feelings on what they had experienced in the workshops. The findings from a sample of one thousand one hundred pupils' questionnaires are presented here. Some of the most frequent words used by the primary school pupils (Year 4, Year 5, and Year 6) to describe the workshops were 'fun', 'liked', 'good', 'interesting', and 'amazing' (see Figure 10). The secondary school (Year 9 pupils) appeared to have a more mature perspective in describing their experience of the workshops. The words used by the secondary school Year 9 pupils were 'think', 'science', 'teach', 'good', 'future', 'help', 'interesting', and 'careers' (see Figure 11). In particular, the Year 9 pupils articulated some constructive impressions of the DTLSN learning community:

- I think it is important for children to learn about computer science because we are becoming more reliable on computers and technology and it will help us in the future. (P1060)
- I think it helps people see more into career paths to do with technology. (P1046)
- I think it's important and good to learn your future strengths and my career options. It was insightful to know and giving me a range of option and paths to take. (P1054)
- I think it is good because it teaches us about technology and potential careers. (P816)
- I think it is very interesting and important that we learn about these things. (P1059)

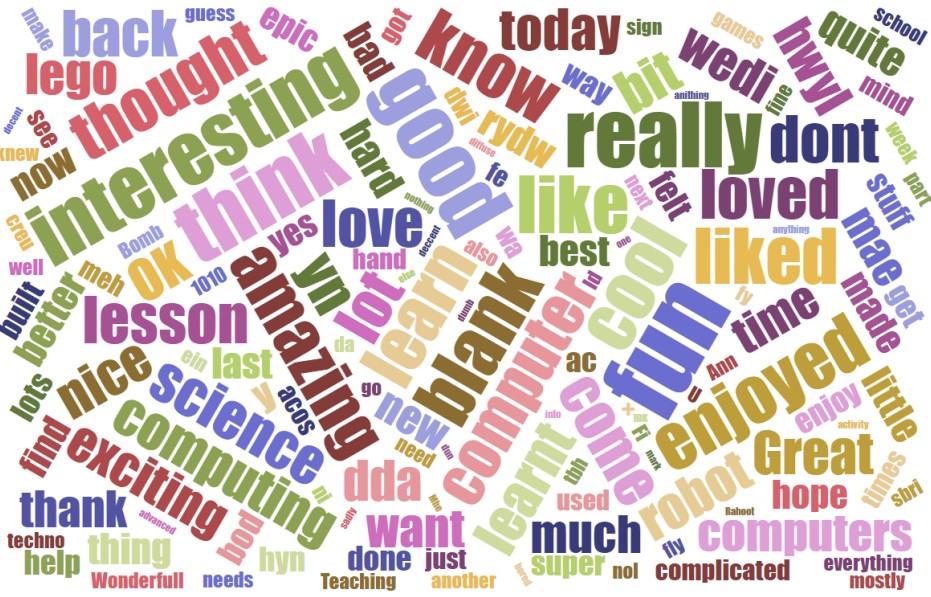

**Figure 10.** Primary school pupil descriptions of workshops.

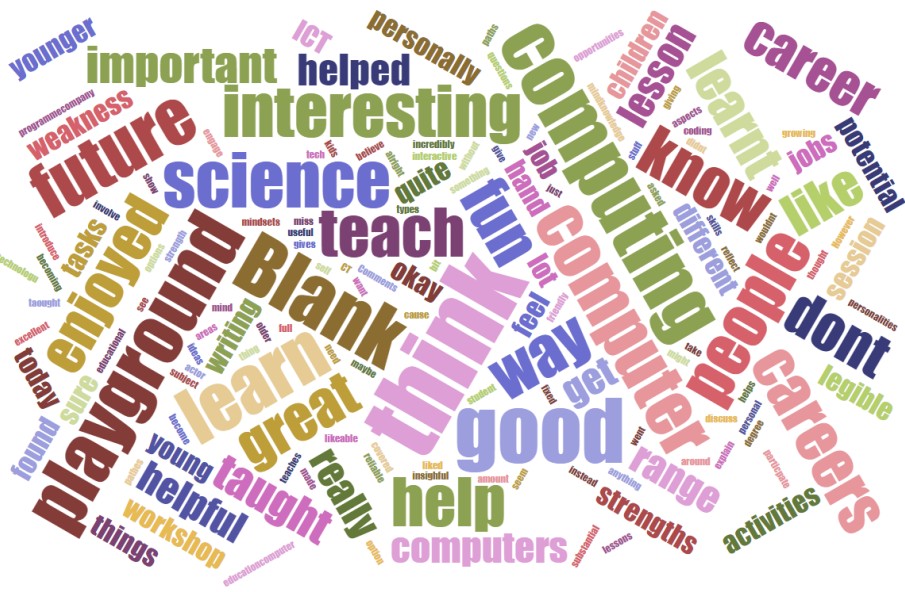

**Figure 11.** Secondary school pupil descriptions of workshops.

Some of the younger participants had a tendency to be confused with some of the workshop materials.

- I think it is kind of confusing but interesting. (P470)
- I like doing fun activities but it gets a bit tricky and confusing. (P504)
- I really enjoyed it but it was confusing too. (P860)

However, the general comments from the primary school pupils showed that they were quite enthusiastic about what they learned. In fact, the most popular workshop was the computational computing workshop, which scored two hundred and ninety-three positive remarks. The Lego Spikes workshop was next, with a score of two hundred and thirty-three remarks, followed by the modelling zombies workshop, which scored two hundred and twenty-four positive remarks. In particular, the primary pupils felt:

- I like doing the butterfly thing. (P539)
- I really liked today's lesson. It was very interesting and fun and it made me curious. (P541)
- I liked the lesson and I hope they come back soon. (P1038)

Figure 12 highlights the interest of participants in taking part in digital technology-based activities. When asked, 'Would you like to learn computer science in your school now?', seven hundred and twenty-eight questionnaires from the one thousand one hundred questionnaires presented 'Yes', sixty-three recorded 'No', and three hundred and nineteen were neutral.

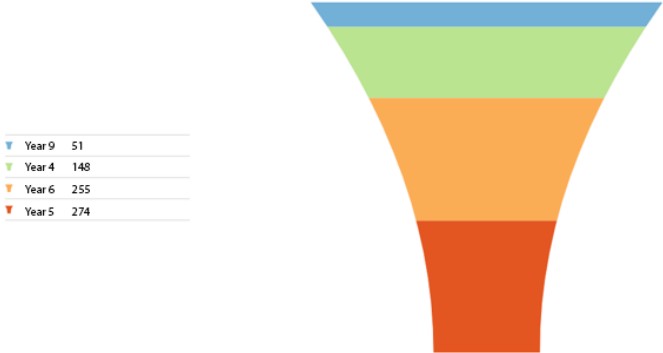

| | Year 9 | 51 |
| | Year 4 | 148 |
| | Year 6 | 255 |
| | Year 5 | 274 |

**Figure 12.** Would you like to learn computer science in your school now?

5.2.4. Discussion

Despite being a short survey, the findings clearly highlight that the DTLSN has started to successfully increase school pupils' awareness (and often knowledge) of digital technology/computer science. Interestingly, it seems that after the workshops, the participants had a more tangible feel (e.g., definite answer) on whether they 'like' or 'dislike' or are 'not sure' about computer science. For example, the computational computing workshop showed two hundred and thirty-six 'likes', eighty-nine 'not sure', and thirty-eight 'Nos'. Furthermore, when asked if they felt that the DTLSN helped them to learn computer science, the majority of participants felt that it had. In detail, nine hundred and sixty-three questionnaires recorded 'Yes', one hundred and forty-one questionnaires recorded 'No', and five reported 'no comment'. After planting the seeds, the next steps for the DTLSN will be to continue to nurture the momentum created by these learning experiences.

*5.3. Reflection*

These studies highlight that the DTLSN learning community has the potential to provide detailed, informative, and enjoyable support for both pupils and teachers in the Blaenau Gwent region, Wales (UK), and beyond. The findings clearly demonstrate that the DTLSN learning community enabled both teachers and pupils to broaden their learning experiences. The pupils particularly enjoyed the computational computing workshop, and it provided the teachers with the means to explore new learning opportunities that could then be further reinforced in other lessons and learning scenarios. However, feedback from the teachers also highlighted some concerns about how the speed of development and change in technology could make it difficult for them to remain up-to-date. As a result, many of the teachers indicated the need for ongoing support from learning communities such as DTLSN if the momentum for the use and learning of technology is to be maintained.

Furthermore, with regard to the four disciplines of the learning partnership ([21]), the DTLSN team realised the importance of the 'convening' discipline. For example, there is a need to actively explore opportunities for funding to maintain the momentum that has been established by this learning community. For the DTLSN learning community, it is crucial that there is a continued and sustainable provision of workshops that promote the development of skills and knowledge for the community. Moreover, upon reflection, there is an opportunity for the DTLSN team to expand the community to other local authority areas and other year groups. The expansion of the community in these ways could allow for more widespread support and leadership as well as skill and knowledge development at a wider level. Following that, there is also an opportunity for DTLSN to open up the learning community to other potential learner groups, particularly those who are at risk of being 'Not in Education, Employment, or Training' (NEET). For example, post-16 workshops could be delivered to community groups, such as refugee and asylum seeker community organisations, fostering communities, and/or young offender institutions. This digital injection into these groups could allow for more grassroots skill and knowledge development, again, at a wider level.

Without a doubt, learning communities have the potential to have a significant impact on the uptake of digital technology skills and awareness of future academic and career opportunities. As seen in the DTLSN learning community, this was achieved by informing the community, industry, teachers, parents, and students about the new curriculum and supporting the development of digital technology within the region. Indeed, developing skills and interest in digital technology can lead to the creation of a skilled workforce, improved job opportunities in digital technology, and the promotion of prosperity for a region/country. It is recognised in the Blaenau Gwent region that without the provision of support for digital technology education, there is a danger that the digital uptake will be slower, fewer opportunities will be created, and the region may fall behind in comparison to other regions. The implementation of DTLSN has also provided greater awareness of the new curriculum for teachers, schools, and parents. It has allowed for the provision of a plan for ongoing support and delivery. Moreover, it has raised the profile of the schools

- involved in DTLSN - as ones with the ability to deliver the important technical aspects of the new curriculum. The DTLSN has also shown schools, specifically teachers, how to utilise resources and engage students in a complex subject in an enjoyable manner. From a teacher's perspective, the impact of the DTLSN learning community on pupils, teachers, and schools is as follows:

For pupils:

- More access to STEM and Digital Competence Framework resources and support.
- Consolidating knowledge and skills in the area of coding and digital technology.
- Raised enjoyment levels.
- Increased levels of critical thinking.
- Increased enjoyment, motivation, and engagement.
- Pupils have more confidence.

For teachers:

- Support in the delivery of the curriculum.
- Consolidating knowledge and skills.
- Beneficial multimodal learning delivery.
- Support gives teachers ideas and confidence to teach new skills to pupils.
- Increased confidence and ability to make links within topics.
- Better understanding of computational thinking.
- Continuous professional development.

For school:

- Great support across multiple-year groups with skill development.
- Deliverers further support during staff meetings.
- Improves school experiences and offers for pupils.
- Increases available resources.

Finally, one shortcoming of this research is the absence of feedback from the families. Despite seven parents engaging in the project, the team was unable (due to tight project turnarounds) to collect data on their thoughts and feelings on the DTLSN. The family is a key stakeholder in the DTLSN and this will be a priority consideration for any future research studies.

## 6. Conclusions

Learning communities, such as the DTLSN, are valuable in how they enable people to connect, inspire, motivate, and learn from one another. They also nurture a sense of community ownership, which can instil confidence. As we have seen in this paper, the DTLSN learning community was strategically designed with a strong communications backbone. Moreover, it took a skills-focused approach to address the gaps in both teachers' and pupils' digital technology learning in the Blaenau Gwent region, Wales (UK). The authors feel that this only can have a positive impact on the Digital Competence Framework (DCF) and the new Curriculum for Wales (2022). Indeed, the DTLSN learning community has demonstrated the ability to provide greater awareness of the subject and, in some instances, has addressed some of the learning and training needs of teachers and school children. As we have heard from the teachers in this project, there is a huge benefit from working together in informal and formal learning communities, such as the Digital Technology Learning Support Network (DTLSN). However, more still needs to be accomplished. For long-term benefits, learning communities, such as the DTLSN, should be effectively primed and funded to play a key role in tackling the digital skill shortages in Wales and elsewhere in the UK. In addition, with the workshop approach delivered by diverse tutors, learning communities, such as the DTLSN, can be designed to support more diversity and address other issues, such as gender inequality in STEM subjects. The future of digital technology learning should be back in the heart of the community. Similar to more traditional literacy skills, parents and teachers need to work together to inspire, motivate, and ultimately equip pupils with digital confidence for their future.

**Author Contributions:** Conceptualization, F.C. and R.F.; methodology, F.C.; validation, F.C.; formal analysis, F.C. and R.F.; writing—original draft preparation, F.C., C.H. and S.M.; writing—review and editing, F.C., S.M. and V.B.; visualization, F.C. and R.F.; project administration, V.B.; funding acquisition, F.C. and C.H. All authors have read and agreed to the published version of the manuscript.

**Funding:** The Research was funded by the UK Community Renewal Fund CRF746109.

**Institutional Review Board Statement:** The study was conducted in accordance with the Declaration of Helsinki, and approved by the Ethics Committee of Cardiff Met University (protocol code CST.Ethics.App.Staff.FCarroll.20220124 and 24.01.2022).

**Informed Consent Statement:** Informed consent was obtained from all subjects involved in the study.

**Data Availability Statement:** Data is unavailable due to ethical restrictions.

**Acknowledgments:** The authors thank Technocamps, Swansea University, for their support and partnership throughout the project, and all of the teachers and pupils who took part in the study. Finally, the authors thank Ciotek, Ltd., for their support in evaluating the project.

**Conflicts of Interest:** The authors declare no conflict of interest.

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
