# Peer review of "The Journey to Making ‘Digital Technology’ Education a Community Learning Venture"

_education, doi:10.3390/educsci13050428_

Round 1

Reviewer 1 Report

Dear Authors  

I appreciate reading your paper. Here are some suggestions for improvement.

In both Study A and B, I suggest replacing the reference to the study duration in the procedure section with information about the data gathering duration.

Figures 8, 9 and 10 should be improved.

The abstract should explicitly mention both studies that were conducted.

Acknowledgment information is missing.

Best regards, 

Author Response

Dear reviewer 1.

Thank you for your feedback. Please see below how we addressed your comments:

Comment 1: In both Study A and B, I suggest replacing the reference to the study duration in the procedure section with information about the data gathering duration.
Amendment: Study A: The study took approximately twenty minutes in duration. The questionnaire was printed out and delivered to the participants at the start of the workshops, to be completed during the workshop. This occurred over a three-month period (April 2022 to June 2022).
Amendment: Study B: The study took approximately ten minutes in duration.  The questionnaire was printed out on one page and delivered to the participants after each of the three workshops. This occurred over a three month period (April 2022 to June 2022).

Comment 2: Figures 8, 9 and 10 should be improved.
Amendment: Figures 8, 9 and 10 have been increased in size so they are now more easily readable.

Comment 3: The abstract should explicitly mention both studies that were conducted.
Amendment: Secondly, through the description, analysis, and interpretation of findings from two studies, the paper will highlight the impact of the DTLSN learning community on 16
teachers and pupils in Blaenau Gwent, especially for their learning and teaching

Comment 4: Acknowledgment information is missing.
Amendment: The authors would like to thank the UK Community Renewal Fund for supporting this project. They would like to thank Technocamps, Swansea University for their support and partnership throughout the project and all the teachers and pupils that took part in the study. Finally, they would like to thank Ciotek ltd for their support in evaluating the project.

Thank you

Fiona

Reviewer 2 Report

The study is purely descriptive. There is a disconnect between the Introduction about CoPs and the actual study. The Methods are inadequate for any empirical analysis, ergo the first comment. The teacher sample is extremely small. The student sample is substantial, but we are told little about them, or the population they might represent. Any information on author bias is missing. In short, the submission sounds like something published in a trade journal, i.e., largely teacher to teacher. If that fits the Journal's mission, then it is okay. As a more rigorous research study, improving the description of Method will not change the nature of the data collected. It is well written, and as a qualitative "case study", it may provide value to a larger practicing community.

Author Response

Dear reviewer 2,

Thank you for your feedback. Please see below how we addressed your comments:

Comment 1: The study is purely descriptive. There is a disconnect between the Introduction about CoPs and the actual study.
Amendment: The last paragraph of the introduction has been reworked to highlight the link between learnings communities and the study.

This paper will present two studies that aim to probe teachers and school pupil's impressions of a new digital learning community (Digital Technology Learning Support Network-DTLSN) in Blaenau Gwent, Wales (UK). The following sections of this paper will discuss the Digital Competence Framework and the new Digital Technology GSCE subject in Wales. It will emphasise the need for a learning community to support teachers and pupils with digital technology. Importantly, the paper will discuss what was required to set up the DTSLN. Through the findings from two studies, it will also highlight the positive impact of such a learning community on teachers and school pupils in their ever-increasing technological driven educational system.

Comment 2: The Methods are inadequate for any empirical analysis, ergo the first comment.

Amendment: The methods section has been revised.
Study A: Methods: A questionnaire was used to gauge teacher’s initial responses to the DTLSN. The questionnaire consisted of twenty-five questions and all teachers were asked to fill it out post workshops. The questionnaire consisted of multiple choice and text-entry type questions. It included questions such as: ‘What are the barriers that you personally have faced in skills development?’ to ‘What impact do you believe DTLSN has had on the following groups-teachers, pupils, school?’.  All teachers were given the option to take part in the research or to opt out.

Study B: Methods: The questionnaire followed a standard format with both open ended and closed questions. It consisted of seven questions mainly multiple choice but with some where the participants had the option to comment. It included questions such as: ‘Did you enjoy the activities and the equipment we used?’ to ‘Are you more interested in computer science/ digital technology now?’. The questionnaire was concisely and simply designed to ensure that all age groups could easily understand what was being asked.

Comment 3:  The teacher sample is extremely small. The student sample is substantial, but we are told little about them, or the population they might represent. Any information on author bias is missing.
Amendment: Additional information has been added here.                             Over five hundred and thirty-one school pupils between the ages of eight years old and fourteen years old took part in this study. These participants partook in all DTLSN workshops which were delivered to twelve schools in the Blaenau Gwent region, Wales, UK in the spring and summer of 2022. To avoid sampling bias, one thousand one hundred participant questionnaires were randomly selected from a larger collection of one thousand five hundred and ninety three participant questionnaires.

Comment 4: In short, the submission sounds like something published in a trade journal, i.e., largely teacher to teacher. If that fits the Journal's mission, then it is okay. As a more rigorous research study, improving the description of Method will not change the nature of the data collected. It is well written, and as a qualitative "case study", it may provide value to a larger practicing community.

Amendment: This has been highlighted in the abstract and introduction:
Abstract: This paper provides a qualitative case study on the development of a new Digital Technology learning community for both primary and secondary school pupils, their teachers and parents in Blaenau Gwent, Wales (UK).
Introduction: The authors of this paper provide a qualitative case study to highlight that learning communities can be an effective way to fill the gap and support teachers, pupils and parents as they try to keep abreast of the new digital technology skills required by them in and out of the classroom.